# New Aptamer/MoS_2_/Ni-Fe LDH Photoelectric Sensor for Bisphenol A Determination

**DOI:** 10.3390/nano12010078

**Published:** 2021-12-28

**Authors:** Hongjie Gao, Yun He, Jiankang Liu

**Affiliations:** The Key Laboratory of Biomedical Information Engineering, Ministry of Education, Mitochondrial Biomedical Research Institute, School of Life Science and Technology, Xi’an Jiaotong University, Xi’an 710049, China; gao317021hongjie@163.com (H.G.); y984420yuner_gao@126.com (Y.H.)

**Keywords:** cardiovascular, bisphenol A (BPA), layered double hydroxide (LDH), photoelectrochemical, MoS_2_/Ni-Fe LDH, aptasensor

## Abstract

Here, a new type of PEC aptamer sensor for bisphenol A (BPA) detection was developed, in which visible-light active MoS_2_/Ni-Fe LDH (layered double hydroxide) heterostructure and aptamer were used as photosensitive materials and biometric elements, respectively. The combination of an appropriate amount of MoS_2_ and Ni-Fe LDH enhances the photocurrent response, thereby promoting the construction of the PEC sensor. Therefore, we used a simple in situ growth method to fabricate a MoS_2_/Ni-Fe LDH sensor to detect the BPA content. The aptasensor based on aptamer/MoS_2_/Ni-Fe LDH displayed a linear range toward a BPA of 0.05–10 to 50–40,000 ng L^−1^, and it has excellent stability, selectivity and reproducibility. In addition, the proposed aptamer sensor is effective in evaluating real water samples, indicating that it has great potential for detecting BPA in real samples.

## 1. Introduction

Cardiovascular disease (CVD) has been the leading cause of mortality and disability-adjusted life years worldwide. Bisphenol A (BPA) is a common endocrine disruptor that is used to manufacture polycarbonate and epoxy resins [1,2]. Manufacturing processes release BPA, which is then introduced into the environment and even food products [3,4]. Studies have revealed the relationship between BPA and the risk of several CVD outcomes and CVD risk factors [5,6]. Low concentrations of BPA could cause CVD, cytotoxicity, genotoxicity and, most severely, cancer [7]. Hence, sensitive testing for BPA is required.

Many analytical techniques have been implemented for the detection of BPA, such as amperometry [8], differential pulse voltammetry [9] and photoelectrochemical aptasensors [10]. Among these methods, some require professional operators and expensive equipment, and some even require laborious processes or severe test conditions. Electrochemical sensors are the most widely used group, due to their simple operation, high sensitivity, rapid response time, cost effective, the potential for miniaturization and the real sample analysis. The photoelectrochemical (PEC) aptasensor has attracted substantial attention through combining photoelectrode with aptamer, due to relatively low cost, miniaturization, fast response, short time and high sensitivity. The PEC aptasensor is usually used as an analytical method for monitoring various targets, including small ions, biomolecules and living cells pollutants. The PEC aptasensing process is related to the photoelectric conversion efficiency, depending on the photoinduced carrier separation and transfer of photoactive material. Due to the introduction of aptamer, the PEC platform not only possesses excellent properties of PEC technique but also obtains high selectivity for target assay. Meanwhile, PEC-based aptasensors are label-free, exempting from complex and arduous process of labeling aptamers which might affect the bio-affinity between aptamers and their target analytes. A novel photoelectrochemical (PEC) aptasensor has been employed to analyze BPA, which provided minimal background signals, high selectivity, minimal cost and wide practicability [11,12]. Thus, the PEC technique is an ideal choice to determine the low level of BPA. Here, what we should note is that selecting an appropriate photoactive material to construct the PEC sensing platform is very critical to achieve high performance for BPA detection.

Layered double hydroxide (LDH)-based materials have been applied in photocatalysis, due to their redox efficiency [13,14]. These materials show no toxicity and can be easily synthesized [15]. New LDH compounds are highly efficient for photocatalytic reactions [16]. They are arranged in the octahedral direction through oxo-bridge linkages. The unsaturated metal cations (M^3+^ and M^2+^) are also widely distributed and tunable, resulting in brucite-like layers [17,18,19]. In addition, the hydroxide ions are coordinated octahedrally. Typically, oxo-bridges promote metal-to-metal charge transfer, serving as important factors for redox reactions under visible light by slowing the recombination of electrons and holes [20].

The peculiar structure of the S-Mo-S atomic interlayer gives MoS_2_ excellent photocatalytic properties [21,22,23]. Recent studies have shown that MoS_2_ displays desirable photocatalytic activity due to its distinct electrical and optical properties and large surface area [24,25,26]. Consequently, MoS_2_ has been widely used to couple materials such as BiVO_4_ [27], WS_2_ [28] and TiO_2_ [29] to enhance electron–hole pair separation and, thus, boost PEC performance.

In this study, MoS_2_/NiFe-LDH nanosheets were synthesized by using a simple in situ growth approach, and a sensitive label-free PEC aptasensor was constructed on MoS_2_/NiFe-LDH nanosheets for the nontoxic measurement of BPA. Because of the nanoheterojunctions, Mo_2_S showed optimal separation of photogenerated charge carriers and also increased the visible-light absorption of NiFe-LDH. It also promoted the PEC conversion efficiency of Mo_2_S/NiFe-LDH. These findings demonstrated that NiFe-LDH modified by Mo_2_S displayed excellent PEC activity, biocompatibility and stability for efficient biomolecule detection.

## 2. Experimental

### 2.1. Reagents and Chemicals

Analytical purity reagents were employed directly. Sinopharm Chemical Reagent Co., Ltd. (Beijing, China) (www.sinoreagent.com (accessed on 2 May 2021)), provided thiourea (CH_4_N_2_S), (NH_4_)_6_Mo_7_O_24_∙4H_2_O, nickel nitrate hydrate (Ni(NO_3_)_2_·6H_2_O), ferric nitrate hydrate (Fe(NO_3_)_3_·9H2O) and ethanol, and XFNANO Materials Tech Co., Ltd. (Nanjing, China), provided glucose. BPA aptamers (5′-CCG GTG GGT GGT CAG GTG GGA TAG CGT TCC GCG TAT GGC CCA GCG CAT CAC GGG TTC GCA CCA-3′) were purchased from Shanghai Sangon Biotechnology Co. Ltd. (Shanghai, China). The 0.1 M phosphate buffer (pH = 7.0) was prepared by mixing the stock solution of Na_2_HPO_4_ and NaH_2_PO_4_.

### 2.2. Material Synthesis

At first, Ni (NO_3_)_2_·6H_2_O (1.75 g) and Fe (NO_3_)_3_·9H_2_O (0.81 g) were dissolved in 20 mL H_2_O with sonication for 0.5 h. The solution was transferred to a 50 mL Teflonlined autoclave and the heated at 150 °C, for 10 h, to obtain NiFe-LDH. Secondly, the MoS_2_/LDH heterojunction was prepared. An appropriate amount of (NH_4_)_6_Mo_7_O_24_∙4H_2_O (1.06, 2.83 and 4.24 mg) was dissolved in the mixed solution (60 mL) of ethylene glycol (EG) and H_2_O (EG:H_2_O = 1:2). Then, appropriate amounts of SC(NH_2_)_2_ (0.91, 2.43 and 3.65 mg) and the synthesized 50 mg LDH were slowly added to the suspension obtained above and continuously stirred for half an hour. The mixed solution was heated at 200 °C for 24 h in a 100 mL Teflon-lined autoclave. Finally, the product was continually washed three times with deionized water and ethanol, and then it was dried at 60 °C for 12 h.

### 2.3. Identification

The crystal structure of the samples was characterized by X-ray diffraction analysis (XRD, Bruker D8 Advance, Bruker AXS, Billerica, MA, USA) with Cu Kα radiation. The morphology of the samples was characterized by Scanning Electron Microscopy (SEM, TESCAN VEGA, Guanqian Technology Shanghai Co., Ltd. Shanghai, China) and Transmission Electron Microscopy (TEM, Talos F200C TEM, Thermo Fisher Scientific Electron Microscope, Beijing, China). The X-ray photoelectron spectra were obtained by X-ray photoelectron spectroscope (XPS, ESCALAB 250Xi, Thermo Fisher Scientific, Waltham, MA, USA). Diffuse reflection spectra (DRS) of the samples were texted by using a UV–Vis spectrophotometer with the reference of BaSO4 (UV1800, Shanghai Oasis, Shanghai, China). All PEC experiments were texted with CHI 660E (Beijing Huake Putian Technology Co., Ltd., Beijing, China) electrochemical workstation with a three-electrode system in 0.1 M PBS electrolyte (pH = 7). Pt wire was used as counter electrode and the reference electrode was saturated calomel electrode (SCE). Xenon lamp (PLS-SXE 300, 100 mW·cm^−2^, λ ≥ 420 nm Beijing Bofeilai Technology Co., Ltd. Beijing, China) was applied as the light source. The indium tin oxide (ITO) glass was used as the working electrode. Electrochemical impedance spectroscopy (EIS) was conducted over a frequency range from 1 to 1,000,000 Hz in 0.1 M PBS (pH = 7).

### 2.4. Fabrication of Modified Electrodes

Prior to each modification, the ITO electrodes (10 mm × 15 mm) were washed separately with water, acetone and ethanol for 5 min, respectively, and naturally dried. Typically, 5 mg catalyst powders (MoS_2_/LDH) were dispersed in 1 mL mixed solution (deionized water and chitosan; the water/chitosan ratio is 2/3) to form homogeneous suspension. Then ITO electrode (0.5 cm^2^) was coated with 20 µL suspensions. Then MoS_2_/LDH electrodes were coated with 20 µL BPA aptamer suspensions (1 µmol L^−1^) by the specific chemisorption between aptamer molecules and oxides and incubated at room temperature for 12 h. Then, they were rinsed adequately with distilled water to remove excess BPA aptamer. The fabricated aptamer/MoS_2_/LDH was dried and used for further studies.

## 3. Results and Discussion

### 3.1. Physical Characterization

XRD was used to investigate the phases of all synthesized materials (Figure 1). The peaks at 11.69°, 23.08°, 33.72°, 34.47°, 38.81°, 45.88°, 59.82° and 61.27° in the pattern of pure NiFe-LDH were indexed to the (003), (006), (101), (012), (015), (018), (110) and (113) NiFe-LDH lattices (JCPDS No. 40-0215). The high purity of the synthesized NiFe-LDH was confirmed by the absence of any other peaks. The maxima at 12.6°, 31.2° and 34.6° were attributed to the MoS_2_ lattice planes (002), (100) and (102) (JCPDS No.37-1492). The peak positions of MoS_2_/LDH were consistent among all composites, with both LDH and MoS_2_ peaks were easily distinguishable. Moreover, as the MoS_2_ content was increased, the (002) peak became increasingly prominent, indicating the successful synthesis of MoS_2_/LDH.

XPS analysis was also conducted to investigate the surface composition and chemical states of MoS_2_/LDH. The doublet peaks centered at 167.7 and 162.3 eV corresponded to S 2p_1/2_ and S 2p_3/2_, confirming the presence of metal sulfides (Figure 2A) [30,31]. As presented in Figure 2B, Ni^2+^ peaks at 855.6 and 873.7 eV were attributed to Ni 2p_3/2_ and Ni 2p_1/2_ [32]. Satellite peaks appeared at 878.9 and 861.7 eV. The peaks at 712.4 and 724.6 eV in the Fe 2p species (Figure 2C) were attributed to the Fe 2p_3/2_ and Fe 2p_1/2_ states of Fe^3+^, which appeared because Fe in NiFe-LDH existed in the 3+ oxidation state [33]. The peaks in the XPS spectrum of Mo 3d_3/2_ at 231.6 eV and Mo 3d_5/2_ at 228.4 eV were attributed to Mo^4+^ in Figure 2D [34,35]. The XPS results confirmed that MoS_2_/NiFe-LDH was fabricated.

SEM was used to evaluate the morphology of MoS_2_, LDH and MoS_2_/LDH-5% composites. Figure 3A shows that LDH is formed by staggered stacking of many layers. Figure 3B shows pure MoS_2_ as coiled and interlaced nanosheets. The microstructure of MoS_2_/LDH-5% is shown in Figure 3C,D. The crystallinity and composition of the as-synthesized material structure were further validated by using high-resolution transmission electron microscopy (HRTEM). Figure 3D displays the HRTEM image of the boxed region. The HRTEM data (Figure 3E) reveal two distinct lattice spacings of 0.62 and 0.252 nm, which are in close agreement with those of MoS_2_ (002) (0.615 nm) and LDH (012) (0.260 nm). Figure 3F–I depicts the distribution of S, Mo, Ni and Fe atoms in MoS_2_/LDH-5%, respectively. All elements were spread evenly across the MoS_2_/LDH-5%, indicating that the synthesis was successful.

UV-Vis DRS spectra were measured for MoS_2_, LDH and MoS_2_/LDH-5% to learn more about their ability to handle visible light. Figure 4A shows that all catalysts displayed a high visible-light absorbance. LDH absorbed strongly at 400 nm. Due to the presence of MoS_2_, MoS_2_/LDH-5% displayed better absorption than LDH. The PL spectrum of the composite was examined to learn more about the charge separation and combination mechanisms of carriers during PEC measurements, as shown in Figure 4B. For the LDH and MoS_2_/LDH-5% composites, a prominent peak appeared at 450 nm, as well as a less intense peak for MoS_2_/LDH. As a result, the charge-carrier recombination efficiency was lowered in the composite made of MoS_2_ and LDH [36,37,38]. Because of these factors, MoS_2_/LDH-5% achieved outstanding PEC results.

### 3.2. Photoelectrochemical Characterization

The photocurrent outputs of various electrodes were investigated at 0.1 V with visible-light stimulation in a 0.1 M phosphate-buffered solution (pH = 7.0) with ascorbic acid (AA, 0.1 M). Figure 5 shows a comparison of the photocurrents of all composites. The photocurrent of the composites increased after different quantities of MoS_2_ were added. In Figure 5A, MoS_2_/LDH-5% showed the maximum photocurrent among MoS_2_/LDH composites, and the photocurrent values followed the order MoS_2_/LDH-5% > LDH > MoS_2_; the photogenerated electrons of MoS_2_ tend to recombine with VB holes of LDH, because the VB band of LDH is higher than that of MoS_2_; AA is oxidized by the holes of MoS_2_. Then, photogenerated electron of LDH can efficiently transferred, thus reducing the charge recombination rate, which indicates that the addition of MoS_2_ to LDH improved the visible-light-harvesting performance, while also efficiently prevented LDH charge carrier recombination; thus, the photoelectrochemical activity was improved (Figure 1) [39,40]. The above findings are in line with the EIS spectra (Appendix A).

Moreover, after BPA-aptamer modification, the photocurrent of aptamer/MoS_2_/LDH-5% decreased to 96% of the MoS_2_/LDH-5% because of the steric hindrance produced by the immobilized aptamer, as well as the electrostatic repulsion between the electronegative AA molecule and the negatively charged phosphate skeleton of the aptamer. Consequently, BPA solution could specifically bind to the aptamer. Since the increased steric hindrance hampered the coupling of AA and photogenerated holes and also encouraged photogenerated hole/electron interactions, the photocurrent was decreased. The above-mentioned moleculars displayed steric resistance at the electrode junction, which inhibited electron transport to the electrode surface and thereby lowered the photocurrent (Figure 5B). The electrostatic repulsion of these substances at the electrode interface prevented electron transport and enhanced electron–hole pair recombination, which lowered the photocurrent. More BPA encouraged the development of aptamer–BPA complexes, which reduced the photocurrent. The BPA concentration can be measured by monitoring the decrease of photocurrent density (Figure 1) [37,41]. The above characteristics indicate that the MoS_2_/LDH-5% PEC aptasensor can be used to detect BPA.

Figure 6A shows how the applied potential was chosen. The aptamer/MoS_2_/LDH-5% was tested under different potentials, from −0.2 to 0.3 V in the dark and under illumination. When the potential was changed from −0.2 to 0.3 V, the current between illumination and the dark reduced abruptly. On the other hand, it declined softly when the potential was changed from −0.2 to 0.3 V; thus, 0.1 V is the most appropriate voltage for PEC. The pH of the electrolyte must be regulated during detection because it could impact PEC performance (Figure 6B). Upon increasing the pH of the electrolyte from 5 to 7, the photocurrent was increased. As the pH increased from 7 to 9, the photocurrent was decreased. The maximum photocurrent was reached at pH = 7, probably because the neutral environment was more conducive to maintaining the activity of the aptamer. The aptamer concentration (0.1–2 μmol/L) was also adjusted (Figure 6C) to obtain the maximum sensitivity. The results demonstrated that the largest photocurrent was obtained at a concentration of 1.0 μmol/L. It has been demonstrated that the optimal aptamer concentration has the capacity to trap BPA molecules’ sensitivity, while redundant immobilized aptamer exhibited a larger steric hindrance effect, which can block the BPA-capture course. Thus, 1.0 μmol/L was employed as an optimal aptamer concentration for the immobilization in the following experiments. The effect of the photocurrent on the BPA binding time, using an aptamer, was investigated. The time required for BPA to bind to the aptamer was changed from 5 to 60 s. Figure 6D shows that the photocurrent decreased over time until it stabilized, suggesting that the aptamer/BPA complex permeated the surface of the electrode.

In a 0.1 M phosphate buffer solution (pH = 7.0) containing ascorbic acid (AA, 0.1 M), the stable current of the aptamer/MoS2/LDH-5% electrode under dark conditions was measured. After 20 s, the visible light is turned on; the visible light is turned off after the photocurrent is increased and stabilized, the corresponding concentration of BPA is added after the dark current is stable and then the visible light is turned on to test the photocurrent at this concentration. After adding BPA, the photocurrent was decreased, as seen in Figure 7A. The linear equations were as follows: △*I* = 0.159 + 0.048*c* (*R*^2^ = 0.9946), with a linear range of 0.05–10 ng L^−1^; and △*I* = 0.696 + 6.19 × 10^−5^*c* (*R*^2^ = 0.9932), ranging from 50 to 40,000 ng L^−1^ (△*I* = *I*_0_ − *I*, where *I*_0_ is the photocurrent before BPA was added, and *I* is the photocurrent after BPA was added). Furthermore, the sensor showed a low detection limit of 0.0052 ng L^−1^ (*S*/*N* = 3). At low BPA levels, the local concentration at the electrode surface was rapidly depleted, as the substrate was converted into a product by the catalysis, resulting in the high sensitivity of the electrode response. At higher BPA concentrations, the nano-material is supplied with substrate for a longer period of time and the reaction proceeds over a larger time window. This, together with the possibility of fouling the electrode surface by the reaction products, results in a lower slope. It also attains the saturation level at a higher concentration. Thus, the sensor showed different linear correlations at different concentration ranges [42], as shown in Appendix A. The comparison shows that the detection limit of the aptamer/MoS_2_/LDH-5% photoelectrode was 0.0052 ng L^−1^ in Appendix A. The detection line of the aptamer/MoS_2_/LDH-5% photoelectrode was lower in area PEC. As a result, the aptasensor featured a large linear range and a low detection limit.

To evaluate the stability of the aptamer/MoS_2_/LDH-5% photoelectrode, it was examined for roughly 600 s (Figure 7C). The photocurrent responses of five aptamer/MoS_2_/LDH-5% electrodes were calculated in parallel (Appendix A), and the stability of the aptamer/MoS_2_/LDH-5% photoelectrode was assessed by measuring the photocurrent sensitivity to 50 ng L^−1^ BPA every two days (Appendix A). The aptamer/MoS_2_/LDH-5% appeared to have excellent compatibility and stability. Anti-interference studies were used to test the selectivity of aptamer/MoS_2_/LDH-5%. The amperometric responses of aptamer/MoS_2_/LDH-5% to 50 ng L^−1^ BPA and interfering compounds, including hexafluorobisphenol A (6F-BPA), catechol (CAT), tetracycline (TC), 4-chlorophenol (4-CP), norfloxacin (NOR), etc., are shown in Figure 7D. The interfering photocurrent was small, indicating that aptamer/MoS_2_/LDH-5% possesses better aptasensor BPA selectivity.

To test the dependability of the aptamer/MoS_2_/LDH-5% photoelectrode, Appendix A, it was used to detect different concentrations of BPA in actual water samples. Shahu provided the river water (Wuhan City, Hubei Province), and boiled and filtered water samples were stored at 4 °C. The BPA sample was added to the buffer solution in the presence of ascorbic acid, and the current difference before and after adding the BPA sample was measured. The corresponding BPA concentration was calculated by bringing the current difference into the BPA calibration curve. The RSD was less than 3.24%, and the recoveries were 98.60–100.53%. These findings imply that the BPA aptasensor based on MoS_2_/LDH-5% can detect BPA in real-world samples.

## 4. Summary

For the selective detection of BPA, a new PEC method based on a MoS_2_/LDH-5% aptasensor was developed. The resulting PEC sensor displayed a wide linear range, low detection, acceptable stability and high selectivity, demonstrating its excellent performance for the detection of BPA in actual samples. This research provides a novel method for using PEC sensors for detecting residues in environmental samples.

## Data Availability

The data presented in this study are available on request from the corresponding author.

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
