# Peer review of "New Aptamer/MoS2/Ni-Fe LDH Photoelectric Sensor for Bisphenol A Determination"

_nanomaterials, 2021, doi:10.3390/nano12010078_

Round 1

Reviewer 1 Report

Bisphenol A (BSA) is an organic monomer used to make common consumer goods which are heavily produced worldwide. Upon entering living organisms BPA, as a potent endocrine disruptor, negatively affects various internal organs and regulatory systems and leads to development of different metabolic diseases, especially cardiovascular diseases. Hence there is need to develop a simple and sensitive methods for BPA detection in environment samples.

Comments to the Article:

  1. It should be noted thirst of all that the Title is completely not appropriate. The proposed sensor just can reveal a BPA concentration, but can not prevent cardiovascular diseases, because it is not a medication or a treatment method.
  2. A detailed description for the real sample analysis procedure has to be added. Namely, BPA calibration plot represents dependence of difference between currents measured before and after BPA addition in the buffer solution in presence of the ascorbic acid. It has to be clear how to reproduce the same measurement conditions for an real sample. Moreover it is desirable to add data describing variability of measurements (dispersion or confidence intervals) to the Table S2
  3. A description of mechanism of signal forming that gives two linear ranges toward BPA with such large difference in sensitivity has to be added.
  4. An mechanism of aptamer adsorption is to be clarified, namely what could affect it (pH, ionic strength and so on)
  5. Data with BPA is to be added to the EIS analysis presented in Figure. S1, as well as description was it performed with/without presence of ferrocyanide.
  6. A procedure and conditions of a regeneration of the sensor between different BPA measurements has to be added.

Question to the authors:

  1. Is there possibility of forming insoluble metal phosphates and soluble Ferrous Ascorbate that could change MoS2/Ni-Fe LDH structure?
  2. It is known that ascorbate can reduce Fe3+ into Fe2+. How this process can influence MoS2/Ni-Fe LDH structure?
  3. Why the second BPA concentration range can not be extended for larger BPA concentrations?
  4. Why during optimization process with pH and aptamer concentration the target was just value of photocurrent? It could be better to make the Relative response as the target to get better analytical parameters of the sensor.

Corrections to the article text:

  1. Page 3, line 97 "Then 20 μL suspensions were coated 97
    on ITO electrode
    ". The sentence in not correct, the better is "ITO electrode was coated with ...". The same is in line 98 "BPA aptamer suspensions (1 μmol L-1) were coated"
  2. Page 7, line 179 "The electrostatic repulsion of these proteins at the electrode interface". What are the proteins mentioned here?
  3. Page 8, lines 198 and 200, Page 9 lines , "The aptamer concentration (0.12 M)" Concentration of the aptamer referenced in M (mol/L) units, but must be in micromoles/L.

Author Response

Response letter

(nanomaterials-1366344)

Dear Editor:

Thank you for your suggestive guidelines. According to your suggestions, I have completely revised this manuscript.

Comments and Suggestions for Authors

Bisphenol A (BPA) is an organic monomer used to make common consumer goods which are heavily produced worldwide. Upon entering living organisms BPA, as a potent endocrine disruptor, negatively affects various internal organs and regulatory systems and leads to development of different metabolic diseases, especially cardiovascular diseases. Hence there is need to develop a simple and sensitive methods for BPA detection in environment samples.

Comments to the Article:

1. It should be noted thirst of all that the Title is completely not appropriate. The proposed sensor just can reveal a BPA concentration, but can not prevent cardiovascular diseases, because it is not a medication or a treatment method.

Response: Thank you for your suggestion, we have changed the Title to “New aptamer/MoS2/Ni-Fe LDH photoelectric sensor for bisphenol A determination”.

2. A detailed description for the real sample analysis procedure has to be added. Namely, BPA calibration plot represents dependence of difference between currents measured before and after BPA addition in the buffer solution in presence of the ascorbic acid. It has to be clear how to reproduce the same measurement conditions for an real sample. Moreover it is desirable to add data describing variability of measurements (dispersion or confidence intervals) to the Table S2

Response: We have add the detailed description for the real sample analysis procedure and data describing variability of measurements to the Table S2. “Record the current difference measured before and after the BPA sample is added to the buffer solution in the presence of ascorbic acid, and calculate the corresponding BPA concentration after the current difference is brought into the BPA calibration plot.” 

3. A description of mechanism of signal forming that gives two linear ranges toward BPA with such large difference in sensitivity has to be added.

Response: We added “At low BPA levels the local concentration at the electrode surface is rapidly depleted as substrate was converted into product by the catalysis, resulting in high sensitivity of the electrode response. At higher BPA concentrations, the nano-material is supplied with substrate for a longer period of time and the reaction proceeds over a larger time window. This together with possibility of fouling the electrode surface by the reaction products, results in a lower slope. Also it attains to the saturation level at higher concentration. Thus, the sensor showed different linear correlations at different concentration ranges [43].” to the section of Photoelectrochemical characterization.

4. An mechanism of aptamer adsorption is to be clarified, namely what could affect it (pH, ionic strength and so on)

Response: We have clarified the mechanism of aptamer adsorption in the Photoelectrochemical characterization, “Moreover, after BPA aptamer modification, the photocurrent of aptamer/MoS2/LDH-5% decreased to 96% of the MoS2/LDH-5% because of the steric hindrance produced by the immobilized aptamer, as well as the electrostatic repulsion between the electronegative AA molecule and the negatively charged phosphate skeleton of the aptamer. Consequently, BPA solution could specifically bind to the aptamer. Since the increased steric hindrance hampered the coupling of AA and photogenerated holes and also encouraged photogenerated hole/electron interactions, the photocurrent was decreased. The above-mentioned moleculars displayed steric resistance at the electrode junction, which inhibited electron transport to the electrode surface and thereby lowered the photocurrent (Fig. 5B). The electrostatic repulsion of these substances at the electrode interface prevented electron transport and enhanced electron-hole pair recombination, which lowered the photocurrent. More BPA encouraged the development of aptamer-BPA complexes, which reduced the photocurrent. The BPA concentration can be measured by monitoring the decrease of photocurrent density (Scheme 1) [41,42]. The above characteristics indicate that the MoS2/LDH-5% PEC aptasensor can be used to detect BPA.”

  And We added “The maximum photocurrent was reached at pH = 7, probably because that the neutral environment was more conducive to maintaining the activity of the aptamer.” and “It has been demonstrated that the optimal aptamer concentration have the capacity to trap BPA molecules sensitivity, while redundant immobilized aptamer exhibited a larger steric hindrance effect which can block the BPA-capture course. Thus, 1.0 μmol/L was employed as an optimal aptamer concentration for the immobilization in the following experiments.” to the section of Photoelectrochemical characterization.

5. Data with BPA is to be added to the EIS analysis presented in Figure. S1, as well as description was it performed with/without presence of ferrocyanide.

Response: We have added the data with BPA in Figure.S1, and the EIS analysis were tested in PBS with Fe(CN)63+/Fe(CN)64+(5 mmol L-1).

6. A procedure and conditions of a regeneration of the sensor between different BPA measurements has to be added.

Response: We added “In a 0.1 M phosphate buffer solution (PH = 7.0) containing ascorbic acid (AA, 0.1 M), the stable current of the aptamer/MoS2/LDH-5% electrode under dark conditions was measured. After 20s, the visible light is turned on, the visible light is turned off after the photocurrent is increased and stabilized, the corresponding concentration of BPA is added after the dark current is stable, and then the visible light is turned on to test the photocurrent at this concentration.” in the section of Photoelectrochemical characterization.

Question to the authors:

1. Is there possibility of forming insoluble metal phosphates and soluble Ferrous Ascorbate that could change MoS2/Ni-Fe LDH structure?

Response: Insoluble metal phosphate and soluble ferrous ascorbate will not change the structure of MoS2/Ni-Fe LDH, NiFe LDH is a compound assembled from a positively charged main layer and interlayer anions through electrostatic interaction and hydrogen bonding. This unique structure not only has a huge specific surface area to generate abundant active sites and facilitates electrolyte adsorption And mass transfer channels. Therefore, its good electron transfer ability and structural stability are guaranteed, cited in “Adv. Mater. 2019, 1903909”

2. It is known that ascorbate can reduce Fe3+ into Fe2+. How this process can influence MoS2/Ni-Fe LDH structure?

Response: NiFe LDH is a compound assembled from a positively charged main layer and interlayer anions through electrostatic interaction and hydrogen bonding. This unique structure not only has a huge specific surface area to generate abundant active sites and facilitates electrolyte adsorption And mass transfer channels. Therefore, its good electron transfer ability and structural stability are guaranteed. Ascorbic acid has a low concentration and preferentially binds to the holes of MoS2. This process will not affect the structure of MoS2/Ni-Fe LDH. 

3. Why the second BPA concentration range can not be extended for larger BPA concentrations?

Response: At a higher BPA concentration, it takes longer to provide a substrate for the nanomaterial, the reaction proceeds in a larger time window, and it also reaches a saturation level at a higher concentration and together with the possibility of reaction products contaminating the electrode surface, so the second BPA concentration range can not be extended for larger BPA concentrations.

4. Why during optimization process with pH and aptamer concentration the target was just value of photocurrent? It could be better to make the Relative response as the target to get better analytical parameters of the sensor.

Response: Photocurrent can optimize the reaction conditions more intuitively. Thanks for your suggestions, we will further study in the future work.

Corrections to the article text:

1. Page 3, line 97 "Then 20 μL suspensions were coated on ITO electrode". The sentence in not correct, the better is "ITO electrode was coated with ...". The same is in line 98 "BPA aptamer suspensions (1 μmol L-1) were coated"

Response: Thanks for your suggestion. We have changed these according to your suggestion.

2. Page 7, line 179 "The electrostatic repulsion of these proteins at the electrode interface". What are the proteins mentioned here?

Response: This is our improper use of the word. We have changed “proteins” to “substances”.

3. Page 8, lines 198 and 200, Page 9 lines , "The aptamer concentration (0.1–2 M)" Concentration of the aptamer referenced in M (mol/L) units, but must be in micromoles/L.

Response: Thanks for your suggestion. We have changed these according to your suggestion.

Reviewer 2 Report

The paper can be published after minor revision reflecting comments inserted as yellow notes into attached pdf of submitted manuscript and suplement

Reviewer 3 Report

The manuscript presented describes the preparation of a composite photosensor for the determination of bisphenol A. The protocol of electrode preparation is a complex multistage procedure, but the authors proved its good reproducibility. The analytical performance of the developed sensor competes with previously reported protocols for bisphenol determination.

Unfortunately, the low quality of the data presentation, the lack of fundamental information, the numerous errors of all kinds, the awkward way of describing the results and the ill-justified far-reaching generalizations / conclusions make the manuscript very difficult to follow. The examples of issues are listed below as noted in the manuscript.

Title   

The manuscript title should be revised to make it more accurate, e.g., New aptamer/MoS2/Ni-Fe LDH photoelectric sensor for bisphenol A determination

Abstract

It is recommended to remove the first sentence. The next fragment, starting from ‘According to.... And Ni-Fe LDH’ should be rewritten to provide the general idea of the developed sensor.

Introduction

In the Introduction, the role of all components should be introduced to the reader. What was the purpose or role of the constituents used? Are there alternative methods available for bisphenol A using fluorescence sensors or fluorescence quenching? Many aptamer sensors for bisphenol A have been reported and reviewed. What are their advantages and disadvantages? What advantages can the new sensor provide?

Referring to the information provided in the Introduction, the aim of the work should be stated.

Experimental

The ‘Testimonial’ should be replaced by ‘Experimental’ or a synonym.

The chemical formula should be written using subscript (H2O not H2O, etc.). Also, the notation of pH needs correction (eg PH = 7). The ‘Reagents of analytic level’ should be replaced with ‘analytical grade reagents’. Instead of ‘UV–vis spectrophotometer’, the name should be ‘UV–Vis spectrophotometer’. The same applies to the UV–Vis spectrum, etc.

The procedure of layered double hydroxide should be more precise. What was the mass ratio of Ni (NO3)2∙6H2O, Fe (NO3)3∙9H2O and water?

Instead of the ‘appropriate amount’ the mass of (NH4)6Mo24∙4H2O,  ethylene glycol and H2O should be provided. The same remark applies to thiourea. The description such as ‘Appropriate amount of SC(NH2)2 and the synthesized LDH were slowly added to the suspension obtained above’ – do not provide the reader with any information.

‘Typically, 5 mg catalyst powders (MoS2/LDH) were dispersed in 1 mL mixed solution (deionized water and chitosan)’  - Do the Authors mean that two separate portions of 5 mg of MoS2 and LDH were used to prepare the dispersion? What was the water/chitosan ratio?

In what way was the aptamer suspension prepared? What solution or buffer was used to prepare the working concentration of aptamer?

In the Identification subsection, the information about techniques applied is provided but the details about used devices (instrument name, type, manufacturer) are missing.

Results and Discussion

According to Figure 2, the XPS spectrum of Ni is presented in Figure 2B rather than in Figure 1B, as stated in the manuscript. The same remark applies to the figure showing the XPS of Fe, known as Figure 1C. Instead of Figure 2A, Figure 2E is mentioned in the text.

There is no obligation, but I still think it would be good to edit the caption of Figure 3 to make it more informative. The labels (A), (B), etc. should be located behind the description of the objects shown in the corresponding panel of the figure. For example: SEM images of LDH (A), MoS2 (B), etc.

The objects visible in Figure 3A do not resemble microspheres. It seems that the image labelled as Figure 3B shows MoS2 and not, as stated in the manuscript, Figure 3A.  Authors are advised to check the captions Figure 3A and 3B for correctness.

‘Fig. 3(D) displays the HRTEM image of the boxed region in Fig. 3(E)’ – No object was boxed in Figure 3D or Figure 3E.

‘The HRTEM data (Fig. 3E) reveal two distinct lattice spacings of 0.62 nm and 0.252 nm, which are in close agreement with those of MoS2 (002) and LDH (012)’ – Could the Authors provide reference data reporting the lattice spacings in MoS2 and LDH?

‘Furthermore, the close contact between MoS2 and LDH confirmed structural development and promoted charge transfer and electron transport. 6,25’ – This sentence is confusing and incomprehensible. The works referred are not helpful in this context.

‘Figure. 4 UV-vis diffuse reflectance spectra (A), PL spectra (B)’ – The legend in Figure 4B is not correct. For the reader’s convenience the Authors are asked to use the same line style/colour for the same material and list them in the same sequence in the legend in both figures.

‘As a result, the charge carrier recombination efficiency was lowered in the composite made of MoS2 and LDH [36-38]. Because of these factors, MoS2/LDH- 5% achieved outstanding PEC results.’ – One set of photoluminescence spectra is definitely not enough to draw such far-reaching conclusions.

 ‘The photocurrent outputs of various electrodes were investigated  … in a 0.1 M phosphate-buffered solution (pH 7.0) with ascorbic acid (AA, 0.1 M)’ – Did all the tested solutions contain ascorbic acid?  The legends in Figure 5B indicate that ascorbic acid was present only in three cases out of six investigated. Ascorbic acid is not included in Figure 5A.

It is recommended to describe the processes shown in Scheme 1 in the section ‘Photoelectrochemical characterisation’ or in the Introduction.

‘Moreover, the photocurrent of aptamer/MoS2/LDH-5% decreased to 3.47 μA cm-1 after BPA aptamer modification’ – There is no point in giving the exact value (3.47 μA cm-1) if no reference value is provided. The change expressed as a percentage is more valuable.

‘Moreover, the photocurrent of aptamer/MoS2/LDH-5% decreased … after BPA aptamer modification because of the steric hindrance produced by the immobilized aptamer, as well as the electrostatic repulsion between the electronegative AA molecule and the negatively charged phosphate skeleton of the aptamer. Consequently, 50 ng L-1 BPA solution could specifically bind to the aptamer’ – Why 50 ng L-1 only? What about the other concentrations, higher or lower than 50? This fragment should unquestionably be rewritten.

‘The above-mentioned biomolecules …’, ‘The electrostatic repulsion of these proteins at the electrode interface’ – Which biomolecule and protein do the Authors refer to here? Aptamers and bisphenol A are neither biomolecules nor proteins.

‘The lower photocurrent could be used to quantitatively detect BPA (Scheme 1) [41,42]. The above features demonstrated the creation of a PEC aptasensor for identifying BPA using the MoS2/LDH-5% nanocomposites’ – The Authors showed that in the presence of BPA the fluorescence is quenched when using the aptamer/MoS2/LDH-5% sensor. If you can get an useful BPA sensor, much remains to be done, so the referred fragment needs to be rewritten.

‘The aptamer concentration (0.1–2 M) was also adjusted (Fig. 6C) …’ – the micromolar concentration was applied (as given in Figure 6C and Experimental part)

‘The comparison shows that the detection limit of the aptamer/MoS2/LDH-5% photoelectrode was 0.02 ng L-1 in Table S1’ – Which LOD value is correct 0.0052 or 0.02 ngL-1? The Authors are asked to phrase the fragment reported in rows 216 - 219 differently.

In Figure 7 D caption, the information about the concentration of interfering compounds was not provided.

‘Shahu provided the water (Wuhan City, Hubei Province), and oiled and filtered water samples water samples were stored at 4 oC’ – More detailed information about the water samples (Table S2, samples 1-5) should be provided (was it tap, river, waste water, etc.?). ‘Oiled water samples’ – Do the Authors misspelled ‘boiled water samples’?

‘The resulting PEC sensor displayed a wide detection limit, low potential, high accuracy, and cohesiveness, demonstrating its excellent performance for the identification of BPA in actual samples’ – The sentence needs to be edited. The proper analytical chemistry terms used for characterisation of the analytical method should be used in the correct way.

Author Response

Response letter

(nanomaterials-1366344)

Dear Editor:

Thank you for your suggestive guidelines. According to your suggestions, I have completely revised this manuscript.

Comments and Suggestions for Authors

The manuscript presented describes the preparation of a composite photosensor for the determination of bisphenol A. The protocol of electrode preparation is a complex multistage procedure, but the authors proved its good reproducibility. The analytical performance of the developed sensor competes with previously reported protocols for bisphenol determination.

Unfortunately, the low quality of the data presentation, the lack of fundamental information, the numerous errors of all kinds, the awkward way of describing the results and the ill-justified far-reaching generalizations/conclusions make the manuscript very difficult to follow. The examples of issues are listed below as noted in the manuscript.

Title   

  1. The manuscript title should be revised to make it more accurate, e.g., New aptamer/MoS2/Ni-Fe LDH photoelectric sensor for bisphenol A determination

Response: We have changed the manuscript title according to your suggestion.

Abstract

  1. It is recommended to remove the first sentence. The next fragment, starting from ‘According to.... And Ni-Fe LDH’ should be rewritten to provide the general idea of the developed sensor.

Response: Thanks for your suggestion. We have changed these according to your suggestion.

Introduction

  1. In the Introduction, the role of all components should be introduced to the reader. What was the purpose or role of the constituents used? Are there alternative methods available for bisphenol A using fluorescence sensors or fluorescence quenching? Many aptamer sensors for bisphenol A have been reported and reviewed. What are their advantages and disadvantages? What advantages can the new sensor provide?

Referring to the information provided in the Introduction, the aim of the work should be stated.

Response: Thanks for your suggestion. We added “Among these methods, some require professional operators, expensive equipment, and some require labourious process or severe test conditions. Eletrochemical sensors are the widely used group due to their simple operation, high sensitivity, rapid response time, cost effective, the potential for miniaturization and the real sample analysis. Pho-toelectrochemical (PEC) aptasensor has attracted substantial attention through com-bining photoelectrode with aptamer, due to relatively low cost, miniaturization, fast response, short time and high sensitivity. PEC aptasensor is usually used as an analyti-cal method for monitoring various targets, including small ions, biomolecules and liv-ing cells pollutants. The PEC aptasensing process is related to the photoelectric conver-sion efficiency, depending on the photoinduced carrier separation and transfer of pho-toactive material. Due to the introduction of aptamer, the PEC platform not only pos-sess excellent properties of PEC technique but also obtained high selectivity for target assay. Meanwhile, PEC-based aptasensors are label-free, exempting from complex and arduous process of labeling aptamers which might affect the bioaffinity between ap-tamers and their target analytes. A novel photoelectrochemical (PEC) aptasensor has been employed to analyze BPA, which provided minimal background signals, high se-lectivity, minimal cost, and wide practicability [11,12]. So the PEC technique is an ide-al choice to determine the low level of BPA. Here, what we should note is that selecting an appropriate photoactive material to construct the PEC sensing platform is very crit-ical to achieve high performance for BPA detection.” in the section of Introduction.

Experimental

  1. The ‘Testimonial’ should be replaced by ‘Experimental’ or a synonym.

Response: We have changed “Testimonial” to “Experimental”.

  1. The chemical formula should be written using subscript (H2O not H2O, etc.). Also, the notation of pH needs correction (eg PH = 7). The ‘Reagents of analytic level’ should be replaced with ‘analytical grade reagents’. Instead of ‘UV–vis spectrophotometer’, the name should be ‘UV–Vis spectrophotometer’. The same applies to the UV–Vis spectrum, etc.

Response: Thanks for your suggestion. We have changed these according to your suggestion.

  1. The procedure of layered double hydroxide should be more precise. What was the mass ratio of Ni (NO3)2∙6H2O, Fe (NO3)3∙9H2O and water?

Response: We added “At first, Ni (NO3)2·6H2O (1.75 g) and Fe (NO3)3·9H2O (0.81 g) were dissolved in 20 mL H2O with sonication for 0.5 h.” to the section of Material synthesis.

  1. Instead of the ‘appropriate amount’ the mass of (NH4)6Mo24∙4H2O,  ethylene glycol and H2O should be provided. The same remark applies to thiourea. The description such as ‘Appropriate amount of SC(NH2)2 and the synthesized LDH were slowly added to the suspension obtained above’ – do not provide the reader with any information.

Response: We have added the information of content to the section of Material synthesis.

  1. ‘Typically, 5 mg catalyst powders (MoS2/LDH) were dispersed in 1 mL mixed solution (deionized water and chitosan)’  - Do the Authors mean that two separate portions of 5 mg of MoS2 and LDH were used to prepare the dispersion? What was the water/chitosan ratio?

Response: Instead of using two separate 5mg of MoS2 and LDH to prepare the dispersion, we used MoS2/LDH as a whole to prepare the dispersion. And We added “the water/chitosan ratio is 2/3” to the Fabrication of modified electrodes.

  1. In what way was the aptamer suspension prepared? What solution or buffer was used to prepare the working concentration of aptamer?

Response: BPA aptamers (5’-CCG GTG GGT GGT CAG GTG GGA TAG CGT TCC GCG TAT GGC CCA GCG CAT CAC GGG TTC GCA CCA-3’) were purchased from Shanghai Sangon Biotechnology Co. Ltd. (Shanghai, China), and used 0.1 M phosphate buffer to prepare the working concentration of aptamer.

  1. In the Identification subsection, the information about techniques applied is provided but the details about used devices (instrument name, type, manufacturer) are missing.

Response: We have added the details about used devices to the section of Identification.

Results and Discussion

  1. According to Figure 2, the XPS spectrum of Ni is presented in Figure 2B rather than in Figure 1B, as stated in the manuscript. The same remark applies to the figure showing the XPS of Fe, known as Figure 1C. Instead of Figure 2A, Figure 2E is mentioned in the text.

Response:  We have changed it according to your suggestion.

  1. There is no obligation, but I still think it would be good to edit the caption of Figure 3 to make it more informative. The labels (A), (B), etc. should be located behind the description of the objects shown in the corresponding panel of the figure. For example: SEM images of LDH (A), MoS2 (B), etc.

Response:  We have changed it according to your suggestion.

  1. The objects visible in Figure 3A do not resemble microspheres. It seems that the image labelled as Figure 3B shows MoS2 and not, as stated in the manuscript, Figure 3A.  Authors are advised to check the captions Figure 3A and 3B for correctness.

Response: Thanks for your suggestions, we have checked it.

  1. ‘Fig. 3(D) displays the HRTEM image of the boxed region in Fig. 3(E)’ – No object was boxed in Figure 3D or Figure 3E.

Response: We have changed it to “Fig. 3(D) displays the HRTEM image of the boxed region”

  1. ‘The HRTEM data (Fig. 3E) reveal two distinct lattice spacings of 0.62 nm and 0.252 nm, which are in close agreement with those of MoS2 (002) and LDH (012)’ – Could the Authors provide reference data reporting the lattice spacings in MoS2 and LDH?

Response: We have added the reference data reporting the lattice spacings in MoS2 and LDH.

  1. ‘Furthermore, the close contact between MoS2 and LDH confirmed structural development and promoted charge transfer and electron transport. 6,25’ – This sentence is confusing and incomprehensible. The works referred are not helpful in this context.

Response: We have deleted it according to your suggestion.

  1. ‘Figure. 4 UV-vis diffuse reflectance spectra (A), PL spectra (B)’ – The legend in Figure 4B is not correct. For the reader’s convenience the Authors are asked to use the same line style/colour for the same material and list them in the same sequence in the legend in both figures.

Response: Thanks for your suggestion. We have changed these according to your suggestion.

  1. ‘As a result, the charge carrier recombination efficiency was lowered in the composite made of MoS2 and LDH [36-38]. Because of these factors, MoS2/LDH- 5% achieved outstanding PEC results.’ – One set of photoluminescence spectra is definitely not enough to draw such far-reaching conclusions.

Response: We found through the UV-Vis DRS spectra that MoS2/LDH-5% displayed better absorption than LDH, further, the charge carrier recombination efficiency of MoS2/LDH-5% was lowered in the composite made of MoS2 and LDH, cited in “Sensors and Actuators B: Chemical, 305,2020, 127449”

  1. ‘The photocurrent outputs of various electrodes were investigated  … in a 0.1 M phosphate-buffered solution (pH 7.0) with ascorbic acid (AA, 0.1 M)’ – Did all the tested solutions contain ascorbic acid?  The legends in Figure 5B indicate that ascorbic acid was present only in three cases out of six investigated. Ascorbic acid is not included in Figure 5A.

Response: Ascorbic acid is not included in Figure 5A. And we change the title of Figure 5 to “Figure. 5 (A) Photocurrent outputs of a variety of electrodes; (B) Photocurrent outputs of a variety of electrodes for BPA detection at 0.1 V vs. SCE before and after introducing AA in PBS.”.

  1. It is recommended to describe the processes shown in Scheme 1 in the section ‘Photoelectrochemical characterisation’ or in the Introduction.

Response: We added “the photogenerated electrons of MoS2 tend to recombine with VB holes of LDH, because the VB band of LDH is higher than that of MoS2, AA is oxidized by the holes of MoS2. Then, photogenerated electron of LDH can efficiently transferred, which reduced the charge recombination rate” to the section of Photoelectrochemical characterisation.

  1. ‘Moreover, the photocurrent of aptamer/MoS2/LDH-5% decreased to 3.47 μA cm-1 after BPA aptamer modification’ – There is no point in giving the exact value (3.47 μA cm-1) if no reference value is provided. The change expressed as a percentage is more valuable.

Response: We have changed it to “Moreover, after BPA aptamer modification, the photocurrent of aptamer/MoS2/LDH-5% decreased to 96% of the MoS2/LDH-5%” according to your suggestion.

  1. ‘Moreover, the photocurrent of aptamer/MoS2/LDH-5% decreased … after BPA aptamer modification because of the steric hindrance produced by the immobilized aptamer, as well as the electrostatic repulsion between the electronegative AA molecule and the negatively charged phosphate skeleton of the aptamer. Consequently, 50 ng L-1 BPA solution could specifically bind to the aptamer’ – Why 50 ng L-1 only? What about the other concentrations, higher or lower than 50? This fragment should unquestionably be rewritten.

Response: The use of 50 ng L-1 is mainly to test the phenomenon of photocurrent changes, and higher or lower than 50ng L-1 will have similar imagination. And wehave rewritten this fragment “Moreover, after BPA aptamer modification, the photocurrent of aptamer/MoS2/LDH-5% decreased to 96% of the MoS2/LDH-5% because of the steric hindrance produced by the immobilized aptamer, as well as the electrostatic repulsion between the electronegative AA molecule and the negatively charged phosphate skeleton of the aptamer. Consequently, BPA solution could specifically bind to the aptamer. Since the increased steric hindrance hampered the coupling of AA and photogenerated holes and also encouraged photogenerated hole/electron interactions, the photocurrent was decreased. The above-mentioned moleculars displayed steric resistance at the electrode junction, which inhibited electron transport to the electrode surface and thereby lowered the photocurrent”.

  1. ‘The above-mentioned biomolecules …’, ‘The electrostatic repulsion of these proteins at the electrode interface’ – Which biomolecule and protein do the Authors refer to here? Aptamers and bisphenol A are neither biomolecules nor proteins.

Response: This is our improper use of the word. We have changed “biomolecules” to “moleculars” and changed “proteins” to “substances”.

  1. ‘The lower photocurrent could be used to quantitatively detect BPA (Scheme 1) [41,42]. The above features demonstrated the creation of a PEC aptasensor for identifying BPA using the MoS2/LDH-5% nanocomposites’ – The Authors showed that in the presence of BPA the fluorescence is quenched when using the aptamer/MoS2/LDH-5% sensor. If you can get an useful BPA sensor, much remains to be done, so the referred fragment needs to be rewritten.

Response: We have rewritten it “The BPA concentration can be measured by monitoring the decrease of photocurrent density (Scheme 1) [41,42]. The above characteristics indicate that the MoS2/LDH-5% PEC aptasensor can be used to detect BPA.”.

  1. ‘The aptamer concentration (0.1–2 M) was also adjusted (Fig. 6C) …’ – the micromolar concentration was applied (as given in Figure 6C and Experimental part)

Response: We have changed these according to your suggestion.

  1. ‘The comparison shows that the detection limit of the aptamer/MoS2/LDH-5% photoelectrode was 0.02 ng L-1 in Table S1’ – Which LOD value is correct 0.0052 or 0.02 ngL-1? The Authors are asked to phrase the fragment reported in rows 216 - 219 differently.

Response: We have changed the “0.02 ngL-1” to “0.0052 ngL-1”, and rewritten this passage of rows 216 - 219.

  1. In Figure 7 D caption, the information about the concentration of interfering compounds was not provided.

Response: We have added the concentration of interfering compounds in Figure 7D.

  1. ‘Shahu provided the water (Wuhan City, Hubei Province), and oiled and filtered water samples water samples were stored at 4 oC’ – More detailed information about the water samples (Table S2, samples 1-5) should be provided (was it tap, river, waste water, etc.?). ‘Oiled water samples’ – Do the Authors misspelled ‘boiled water samples’?

Response: Thank you for your suggestion, we have changed it according to your suggestion.

  1. ‘The resulting PEC sensor displayed a wide detection limit, low potential, high accuracy, and cohesiveness, demonstrating its excellent performance for the identification of BPA in actual samples’ – The sentence needs to be edited. The proper analytical chemistry terms used for characterisation of the analytical method should be used in the correct way.

Response: We changed it to “The resulting PEC sensor displayed a wide linear range, low detection, acceptable stability and high selectivity, demonstrating its excellent performance for the detection of BPA in actual samples.” to the section of Summary.

Round 2

Reviewer 1 Report

Thank you for the answers

Author Response

Sincere thanks for your help and guidance,thank you,and wish you success in work.

Reviewer 3 Report

The authors made significant changes to their manuscript. Unfortunately, it still needs some changes. I have listed the details below.

Abstract:

Lines 10 -16 (highlighted version)

The fragment needs correction, especially the new sentences added in the revised version.

Line 106:

The simples were performed by X-ray diffraction (XRD) diffractometer … – The sentence needs correction

Lines 312 -314:

‘Record the current difference measured before and after the BPA sample is added to the buffer solution in the presence of ascorbic acid, and calculate the corresponding BPA concentration after the current difference is brought into the BPA calibration plot’ – sentence needs correction

Author Response

Dear Editor:

Thank you for your suggestive guidelines. According to your suggestions, I have completely revised this manuscript.

Comments and Suggestions for Authors

The authors made significant changes to their manuscript. Unfortunately, it still needs some changes. I have listed the details below.

Abstract:

Lines 10 -16 (highlighted version)

The fragment needs correction, especially the new sentences added in the revised version.

Response: We have changed it according for your suggestion. “Here, a new type of PEC aptamer sensor for bisphenol A (BPA) detection was developed, in which visible-light active MoS2/Ni-Fe LDH (Layered Double Hydroxide) heterostructure and aptamer were used as photosensitive materials and biometric elements, respectively. The combination of an appropriate amount of MoS2 and Ni-Fe LDH enhances the photocurrent response, thereby promoting the construction of the PEC sensor. Therefore, we used a simple in-situ growth method to fabricate a MoS2/Ni-Fe LDH sensor to detect the BPA content. The aptasensor based on Aptamer/MoS2/Ni-Fe LDH displayed a linear range toward BPA of 0.05–10 ng L-1 to 50–40000 ng L-1, and has excellent stability, selectivity, and reproducibility. In addition, the proposed aptamer sensor is effective in evaluating real water samples, indicating that it has great potential for detecting BPA in real samples.”

Line 106:

The simples were performed by X-ray diffraction (XRD) diffractometer … – The sentence needs correction

Response: We have changed it to “The crystal structure of the samples is characterized by X-ray diffraction analysis (XRD, Bruker D8 Advance, Bruker AXS) with Cu Kα radiation.”

Lines 312 -314:

‘Record the current difference measured before and after the BPA sample is added to the buffer solution in the presence of ascorbic acid, and calculate the corresponding BPA concentration after the current difference is brought into the BPA calibration plot’ – sentence needs correction

Response: We have changed it to “The BPA sample is added to the buffer solution in the presence of ascorbic acid, and the current difference before and after adding the BPA sample is measured. The corresponding BPA concentration is calculated by bringing the current difference into the BPA calibration curve.”
